# Case-Based and Quantum Classification for ERP-Based Brain–Computer Interfaces

**DOI:** 10.3390/brainsci13020303

**Published:** 2023-02-10

**Authors:** Grégoire H. Cattan, Alexandre Quemy

**Affiliations:** 1Data and AI, IBM, 30-150 Kraków, Poland; 2Faculty of Computer Sciences, Poznań University of Technology, 60-965 Poznań, Poland

**Keywords:** electroencephalography (EEG), brain–computer interface (BCI), P300, unstructured classification, hypergraph case-based reasoning classifier, quantum classification, variational quantum classifier, quantum-enhanced support vector classifier

## Abstract

Low transfer rates are a major bottleneck for brain–computer interfaces based on electroencephalography (EEG). This problem has led to the development of more robust and accurate classifiers. In this study, we investigated the performance of variational quantum, quantum-enhanced support vector, and hypergraph case-based reasoning classifiers in the binary classification of EEG data from a P300 experiment. On the one hand, quantum classification is a promising technology to reduce computational time and improve learning outcomes. On the other hand, case-based reasoning has an excellent potential to simplify the preprocessing steps of EEG analysis. We found that the balanced training (prediction) accuracy of each of these three classifiers was 56.95 (51.83), 83.17 (50.25), and 71.10% (52.04%), respectively. In addition, case-based reasoning performed significantly lower with a simplified (49.78%) preprocessing pipeline. These results demonstrated that all classifiers were able to learn from the data and that quantum classification of EEG data was implementable; however, more research is required to enable a greater prediction accuracy because none of the classifiers were able to generalize from the data. This could be achieved by improving the configuration of the quantum classifiers (e.g., increasing the number of shots) and increasing the number of trials for hypergraph case-based reasoning classifiers through transfer learning.

## 1. Introduction

Event-related potentials (ERPs) are small potentials elicited by the brain after the onset of stimulation. ERP-based brain–computer interfaces (BCIs) detect ERPs in an ongoing electroencephalogram in response to a planned stimulation, thus detecting the user’s intention to interact with a proposed action. This type of interface, which was imagined in the early 1970s by Vidal [1], allows disabled people to interact with a computer because such interfaces do not require any muscular interaction (e.g., see [2]). An ERP-based BCI consists of the presentation of an expected yet unpredictable and highly distinct stimulus on a screen, such as the oddball paradigm. Other paradigms exist such as motor imagery, in which the user must imagine an action, or steady-state visually evoked potentials, in which stimuli that flicker at different frequencies are presented (e.g., see [3,4]). BCIs based on the oddball paradigm offer a good trade-off between performance and user fatigue (notably visually). Although we focus here on the classification of noninvasive electroencephalography (EEG) data under the oddball paradigm, it should be noted that classification techniques are similar across all paradigms.

BCI performance remains limited for three reasons. First, in the case of scalp EEGs, the electrodes record a low-amplitude signal due to the skull and remain extremely sensitive to electromagnetic fields (e.g., power lines) and muscular artifacts (e.g., movement). Second, a phenomenon exists that is known as BCI illiteracy, which refers to the fact that between 15% and 30% of people are unable to use BCIs. BCI illiteracy has been suggested to be a physiological trait of participants (e.g., see [5]), although recent examples in the literature contested this by pointing to a default in the stimulation process [6,7]. Third, the information transfer rate of BCIs is very low (a writing speed of approximately five words per minute) [2]. Due to their low information transfer rate compared with mechanical inputs, BCIs are best applied in the clinical domain (e.g., see [8]), where they offer a valuable alternative to people with motor disabilities, such as patients suffering from locked-in syndrome. However, beyond their potential to improve communication in the clinical domain, BCIs based on error-related potentials are useful in movement-trajectory correction for people with disabilities [9,10]. Experiments that integrated BCIs for general public use mostly mitigated the low information transfer rate by implementing a dedicated design considered these interfaces’ lack of reliability [11,12,13,14].

To improve BCIs’ performance, much effort has been invested into research on reliable classifiers such as discriminant analysis, support vector machine (SVM) classifiers, neural networks [15], random forests [16], and Riemannian geometry [17]. Note that the term “neural network” regroups a large set of models ranging from convolutional and deep learning networks to spiking neural networks [18] and the Squeeze-and-Excitation Network with Domain Adversarial Learning (SEN-DAL) [19]. This late model focuses specifically on the classification of multi-modal physiological signals, which has been investigated as a way to improve performance by exploiting the complementary information of the signals (e.g., see [20]). Neural networks are known to perform well with minimally pretreated raw data, whereas other type of models require the use of EEG preprocessing technics such as denoising or artifact rejections (e.g., see [21,22,23]). In particular, [23] illustrated how eye blink detection can be used at the same time for artifact rejection and as a second modality to improve the classification of EEG signals.

To the best of our knowledge, approaches that rely on Riemannian geometry, such as mean-distance-to-mean (MDM) classifiers, have achieved the best performances during international competitions with reported accuracies close to 90% within a couple of seconds [17,24,25]. To complement these studies, we investigated the use of hypergraph case-based reasoning (HCBR) [26] as well as classifiers based on quantum computers for EEG-based BCIs. Two types of quantum classifiers were considered in this study: variational quantum classifiers (VQCs) and quantum-enhanced SVM classifiers (QSVC). On the one hand, HCBR is a classifier designed for unstructured space that may help to reduce the complex preprocessing steps required for BCI classification. On the other hand, encouraging results have been reported for quantum classifiers—particularly QSVC—when compared with traditional classification; such classifiers showed possible advantages in computation time and learning outcomes [27,28,29]. These advantages, along with the development of cloud-based quantum computer services (e.g., the IBM quantum experience by IBM, Armonk, NY, USA) and the continuous improvements in quantum volume, have made quantum classification a promising alternative to classical computing.

In this study, we measured the balanced accuracy of VQCs, QSVCs, and HCBR classifiers and then compared them with state-of-the-art classifiers—namely SVM and MDM classifiers. We assessed the performance of HCBR using both direct and preprocessed data. The results revealed that all of the classifiers were able to learn from the data, and the best training accuracies were obtained using HCBR (71.10%) and QSVC (83.17%). However, more research is required to improve the prediction accuracy of both quantum and case-based classifiers.

The remainder of this paper is structured as follows. Section 2 presents the classification algorithms, Section 3 describes the data that we used for the analysis, Section 4 describes the methods that we employed to assess the performance of HCBR and the quantum classifiers, and Section 5 presents the results of the experiments. Finally, Section 6 provides a discussion and the conclusions of the study.

## 2. Algorithms

### 2.1. Hypergraph Case-Based Reasoning

The HCBR method is dedicated to binary classification and can work in spaces without metrics. It is a Bayesian method of learning metrics. In particular, it was designed to work with any type of data (e.g., textual and digital) and to not be sensitive to missing values.

HCBR models a dataset as a hypergraph in which each vector (called “cases” because they are not vectors per se (the dimensions of cases could vary within the same dataset because data were missing or because some cases consisted of more information)) is an edge. The partition of cases forms a tribe on which probabilistic modeling is based (Figure 1). This partition is transformed into an interaction graph through a projection operation. Intuitively, the construction of the model attempts to answer the following questions: How important is an element of the partition compared with a case? What is the potential for analogy and counterexamples between one case and the other cases containing the element e?

Unlike most metric learning methods, HCBR does not look for a single matrix or vector but rather for both a vector *μ,* which represents the “intrinsic strength” of the partition elements; and a matrix *W*, which represents the importance of each element in the partition for each case:(1)s=WμW∈M(ℝ)N×M,μ∈ℝM∥wj:∥1=1∀1≤j≤N∥μ∥1=1

Starting from an a priori probability for *μ,* an iterative algorithm will adjust *W* and *μ* to increase the consistency of all decisions. The prediction for a new case is made by projecting the case onto the hypergraph and calculating its support, which is defined as the convex combination between *μ* weighted by the proportion of intersections with the elements of the partition. This is a local method in the sense that if the new case does not intersect the hypergraph, then the model cannot make a decision.

It has been demonstrated that the training algorithm converges in a finite time and that a model built using HCBR converges in law toward the underlying distribution with the number of examples in the training game, thereby indicating that the modeling is similar to a mixed model. Finally, if theoretically the temporal complexity of the construction of the hypergraph is exponential, it is demonstrated that in practice an upper bound is given by the cardinality of the number of bits in the dataset. Training and prediction are respectively linear in the number of cases and iterations as well as the cardinality of the case to be predicted.

HCBR was initially developed to be applied in the judicial field; that is, with the constraint of users having limited expertise in data science and the strong constraint of having to justify each decision. The graph structure makes it possible to immediately justify a case using both the importance of the elements that compose it and their respective support of one class or another as well as through quotations by analogy and counterexamples with the immediately adjacent cases—as lawyers could do in a trial. In practice, without hyperparameter optimization, HCBR was the most robust method (resp. second) used on seven different datasets with regard to accuracy (resp. Matthew’s correlation coefficient). It also greatly improved the results of the prediction of decisions of the European Court of Human Rights compared with a baseline study [30].

### 2.2. Foundation of Quantum Computation

When it comes to physics, Newtonian mechanics fail to describe the behavior of particles such as photons or electrons. An example of this assertion is the double-slit experiment (Figure 2) that is commonly used to illustrate the wave–particle duality of light. In 1802, Thomas Young showed that by presenting two slits in front of a light ray, an interference pattern is produced, thus suggesting that the light is like a wave. Interestingly, further double-slit experiments demonstrated that when placing a detector at one of the slits, the interference pattern disappeared, and that each detected particle only passed through one slit in the manner in which a classical particle would have behaved. This illustrated that: (1) the light behaved as a wave and a particle simultaneously, and (2) the results of the experiments also depended on the observer.

From a computational perspective, this suggests that in an enclosed system, one particle acts as if it is passing through the two slits at the same time. In other words, a single bit of information is encoded into a quantum bit that is in fact the superposition of two different states. A measure operation (here a removable projection screen) then translates the result of these simultaneous computations (interference pattern) for the sake of a human observer.

Since a particle is also described by its wave function, the reduction of this superposition of states to a mixture of states is often referred to as the *wave function collapse*. Quantum decoherence [31] provides an explanatory mechanism for the appearance of this phenomenon by demonstrating that multiple interactions with the environment may cause a shift in the wave phases, which then tend to become orthogonal. Whether this collapse occurs is an interpretational problem. In the many-worlds theory [32], for example, all possible outcomes are realized and there are no wave function collapses.

In 1994, Shor demonstrated that it was possible to solve the integer factorization problem in a polynomial time using a quantum algorithm versus an exponential time using a classical approach [33], thereby exposing a potential treat for public-key cryptography schemes. Outside the trending topic of quantum cybersecurity [34], quantum computation is a promising technology that can be used to solve optimization and machine learning problems in areas such as chemistry [35], routing [36], or biology [37]. The enablement of specific tool chains to build quantum applications in a cloud environment is a major factor in furthering the acceptance of quantum computing [38]. Another challenge is the design of quantum computers with low noise and decoherence at room temperature [39].

### 2.3. SVM-like Quantum Classification

In the present study, we used the quantum SVM implementation presented in [40] and contained within the Qiskit library [41]. Here, for the sake of clarity, we resume with some contextual elements explained in [40] regarding SVM quantum classification.

The first concern regarding quantum classifiers is the encoding of classical data in quantum states. This operation is known as feature mapping. To obtain an advantage over classical computing, feature mapping must implement quantum circuits, which are difficult to emulate on a classical computer.

Feature mapping is common to VQCs and QSVCs. Both are SVM-like classifiers in the sense that they generate a separating hyperplane. The difference between them is that VQCs use a variational quantum circuit (also known as a variational form) for this task, whereas QSVCs use a quantum-enhanced but conventional SVM. Quantum computation inside QSVCs occurs at two moments: when estimating the kernel for all pairs of training data and when estimating the kernel for a new datum. For VQCs, this is a four-step process that can be described as follows (Figure 3):

(1).A reference quantum vector ${|0\rangle }^{n}\;$is associated with the data $\vec{x}\;\epsilon \;Ω$ using a nonlinear feature map.(2).A discriminator that corresponds to a short-depth circuit with one to four layers is applied to the data. Short-depth circuits are algorithms that are suitable for error-mitigation techniques because quantum decoherence increases with the depth of the circuit (e.g., see [42]).(3).The output of the discriminator circuit is measured and mapped to a label $y\;\in \left\{-1;1\right\}$that corresponds to the class of the binary classifier.(4).An empirical distribution ${\hat{p}}_{y}\left(\vec{x}\right)\;$ is generated by repeating steps 1 to 3 *R* times (where *R* is the number of shots). Then, labels are assigned according to whether ${\hat{p}}_{y}\left(\vec{x}\right)>{\hat{p}}_{-y}\left(\vec{x}\right)-yb$ (where *b* is a bias parameter).(5).The circuit becomes a binary classifier after the convergence of the algorithm, which determines the correct weights for the discriminator circuit as well as the bias parameter.

### 2.4. Complexity of Quantum SVMs

The complexity of SVMs depends on the kernel function and whether the training or prediction time is being considered. As a reminder, quantum enhanced SVMs are classical SVMs with a quantum kernel. 

At training time, kernel SVMs determine the support vectors. For classical kernels, this task can be restated as a quadratic problem that then can be solved in a polynomial time (e.g., see [43]). For instance, LibSVM [44], a ubiquitous library for SVM implementation that is a base for the popular framework scikit-learn [45], has a time complexity between $O\left(n^{2}\right)$ and $O\left(n^{3}\right)$ depending on the kernel function. In contrast to the classical case, the quantum kernel function cannot be computed exactly because it relies on a fault-tolerant quantum computer. By employing a generalization of the Pegasos algorithm [46], training is then achievable in $O\left(min\left\{M^{2}\varepsilon^{-6},\varepsilon^{-10}\right\}\right)$ (where *M* denotes the size of the data set and *ε* the solution accuracy) [47].

At test time, the complexity of SVMs based on a nonlinear kernel is in general linear based on the number of support vectors $N_{sv}$ and the dimension of the features *d*; that is, $O\left(N_{sv}d\right)$. However, the exact prediction complexity of an SVM based on a quantum kernel is not clear.

In practice, experiments using artificial datasets suggest that quantum-enhanced SVMs offer a provable acceleration compared to classical algorithms [48]. In this respect, the classification of real time-series data within this article is rather novel and original, although the remainder of this paper admittedly focuses on quantum simulation.

## 3. Data

We used data recorded at GIPSA-lab (Saint-Martin-d’Hères, France) that are freely available on Zenodo (Geneve, Switzerland) at https://zenodo.org/record/2649069 (accessed on 20 December 2022). The dataset contained the (noninvasive) EEG recordings of 26 participants (including 7 females) with a mean (SD) age of 24.4 (2.76) who participated in a visual P300 TARGET/NON-TARGET experiment. The visual P300 is an endogenous ERP that peaks at 240–600 ms after the appearance of a visual stimulation on the screen. Unlike short-latency exogenous components, which are automated and sensory responses to a stimulation, endogenous components reflect a neural processing that is solely task-dependent [49]. In particular, the P300 is elicited by the appearance of an improbable and highly distinct stimulation (i.e., the oddball paradigm).

The participants played *Brain Invaders*, a BCI version of the famous vintage game *Space Invaders* (Taito, Tokyo, Japan), consisting of 36 aliens displayed in a 6 × 6 matrix. The participants’ task consisted of counting the number of flashes of a TARGET alien that was designated at the beginning of each set of eight repetitions. In the Brain Invaders P300 paradigm, a repetition was composed of 12 flashes, of which 2 included the TARGET alien and 10 did not (NON-TARGET; see Figure 4). For each participant, there were a total of eight randomly predefined TARGET aliens. Therefore, a total of (resp.) 128 (8 × 8 × 2) and 640 (8 × 8 × 10) TARGET and NON-TARGET trials were recorded for each participant during the experiment.

EEG signals were acquired using a NeXus-32 biofeedback system (MindMedia, Herten, Germany) equipped with 16 wet electrodes placed according to the 10–20 international system (Figure 5). Signals were recorded at a sampling frequency of 128 Hz. A complete description of the dataset is available in [50].

## 4. Method

Data were filtered between 1 and 24 Hz using a zero-phase IR filter with a hamming window. We extracted all TARGET (*n* = 128) and NON-TARGET (*n* = 640) epochs starting from 100 ms to 700 ms after the onset of stimulation and applied a spatial filter using xDAWN (number of filters = 1) [51]. The use of a spatial filter allowed us to reduce the epochs’ dimensionality (and therefore the computation time) while improving the signal-to-noise ratio. Super-trials were obtained by concatenating to each epoch (X) the temporal prototype of the TARGET response (${\bar{X}}_{TARGET}$). We then computed the correlation matrix of each super-trial [24]. All correlation matrices were symmetric positive matrices (SPD) with a size of 4 × 4 and the following structure:(2)$$\left(\begin{array}{cc} {\bar{X}}_{TARGET}{\bar{X}}_{TARGET}{}^{T} & {\bar{X}}_{TARGET}X^{T}\\ X{\bar{X}}_{TARGET}{}^{T} & XX^{T} \end{array}\right)$$

In consideration of the Riemannian geometry of SPD matrices, we vectorized these matrices via projection into the tangent space of the Riemannian manifold [52]. All vectors contained 10 elements.

The resulting vectors were provided as input to the HCBR classifier, VQC, and QSVC. Since HCBR’s only assumption is that values are indexable and it does not use any metric on the input vector space, very close values were considered potentially different. To increase the occurrence of cases, vectors for HCBR were rounded to the first four decimal digits, and correlation matrices were preferred over covariance matrices because they contained normalized inputs.

Due to the extended computation time with both quantum and case-based classifiers, we excluded a systematic approach (e.g., using GridSearchCV from scikit-learn) to determine the best set of parameters for our classifiers. Instead, we used an exploratory subject and determined the hyperparameters of the classifiers using successive experimentation. Therefore, the configuration of the HCBR classifier was tested on a single subject before running the classification task on the entire dataset (Table 1). This dry run revealed that a higher number of digits diminished the performance of the classifier, while a lower number of digits caused convergence problems. All hyperparameters for HCBR remained untouched and equal to zero except *l*0, which was set to 1. For its part, VQC was run on a two-local parametrized circuit (rotation_blocks = ry, rz; entanglement_blocks = cz; reps = 3; entanglement = full) and optimized using simultaneous perturbation stochastic approximation (SPSA; *c*0 = 4, maxiter = 40). SPSA [53,54] is an optimizer that is suitable for noisy environments, which is the case in a quantum computer [40]. For both the VQC and the QSVC, data were linearly entangled using a second-order Pauli-Z evolution circuit (the so-called ZZFeatureMap in Qiskit). The number of repetitions for the ZZFeatureMap was set to 2, and the number of shots was set to 1024. The quantum computer was emulated using QasmSimulator [41], and we performed 100 runs on an 16-qubit IBM processor available in the cloud [55] to compare the execution time. Hyperparameters for the quantum classifiers were determined using the same subject and trial-and-error approach as for HCBR.

The performance of the three classifiers was assessed via five-fold cross validation and measured using the balanced accuracy metric, which was defined as $\frac{1}{2}\;\left(\frac{A}{A+B}+\frac{C}{C+D}\right)$ (where A and B (resp. C and D) stand for the number of correctly and incorrectly classified NON-TARGET (resp. TARGET) epochs).

To test the impact of the processing steps on performance using HCBR, two other runs were performed using vectorized epochs and correlation matrices. To achieve convergence, redundant information between the correlation matrices was removed. The redundant information consisted of the upper-right block (${\bar{X}}_{TARGET}{\bar{X}}_{TARGET}{}^{T})$, which was solely dependent on the temporal prototype of the TARGET response, and the diagonal of the correlation matrices (since the diagonal of the correlation matrices always contained ones) [24] as illustrated in Figure 6.

The balanced accuracies of the VQC, QSVC, and HCBR classifier were compared with those of state-of-the-art SVM and MDM classifiers. To ensure an objective comparison, the performance of the two classifiers was measured with balanced accuracy using the same dataset. Preprocessing and cross-validation were also similar to those of the other classifiers.

Signal processing was performed using MNE [56], and pyRiemann [57,58] was used for the spatial filtering and manipulation of covariance matrices with Riemann geometry. The code libraries for the HCBR and quantum classifiers can be found in [59] and [41], respectively. The assessment of the classifiers was performed using scikit-learn [45].

## 5. Results

The training accuracies of the HCBR classifier, VQC, and QSVC are displayed in Figure 7.

As indicated in Figure 7, all three classifiers had a training accuracy greater than 0.5 and thus were able to learn from the dataset. The training accuracies for HCBR (71.10) and the QSVC (83.17) were excellent and similar to those usually obtained using the SVM classifier during prediction. This section presents the results obtained during prediction for HCBR, the VQC, and the QSVC.

HCBR obtained a mean (SD) balanced accuracy of 52.04% (3.1%) for the classification of correlation matrices vectorized into the tangent space. The assessment of a single fold (training plus prediction) required approximately 15 min to complete within a single-threaded process run on a computer equipped with an i7 (2.6 GHz) processor and 32 GB of RAM. The classification of preprocessed epochs of the signal (i.e., the correlation matrices projected into the tangent space) achieved slightly better results compared with those of vectorized correlation matrices and raw epochs (direct classification) but with higher inter-subject variability (Figure 8). However, in all cases the balanced accuracy was between 0.4 and 0.6, which indicated that both the TARGET and NON-TARGET vectors were generally classified into the same class independently of the preprocessing steps.

The mean (SD) balanced accuracy for the VQC and the QSVC was 51.83% (2.78%) and 50.25% (0.83%), respectively.

The significance threshold of the dataset was individually estimated for each participant (640 NON-TARGET + 128 TARGET epochs) using permutation testing [60]. We used the implementation of sci-kit learn (permutation_test_score) with 10,000 permutations and a simple classification pipeline based on xDAWN covariance matrices and the MDM classifier. The cross-validation splitting strategy of the permutation test was parametrized with five stratified folds. Then, the significance threshold was defined as the 0.975 quantile (α = 0.05) of the distribution of the test sets’ balanced accuracies outputted by the permutation test [61]. Note that (1) the significance level of a dataset did not depend on the type of the classifier as long it was able to generalize and that (2) the exchangeability condition for the permutation test was guaranteed by design because the eight aliens were randomly defined before the experiment. The significance threshold of the dataset varied between 53–54% for all subjects in the dataset, meaning that all of the proposed classifiers were not able to separate the test data in this dataset with the preprocessing we suggested.

The assessment of a single fold required approximately 8 h to complete on our local machine, whereas it required approximately 2 min on a remote quantum backend. This was compared with the classification time achieved using SVM and MDM, which terminated in less than 2 s.

As depicted in Figure 9, the prediction time of HCBR (764.48 s) was three orders of magnitude larger than the one obtained with the SVM classifier (0.91 s). The quantum classifiers also required a large prediction time (6238.61 s and 39,630.02 s for VQC and QSVC. respectively) with a high standard deviation for QSVC (10,736.88 s). However, the algorithms were emulated on our local machine; hence, future results may vary depending on the machine configuration and whether a quantum backend or a quantum machine emulated on a classical computer is used.

In addition, Figure 10 illustrates how the accuracies obtained with quantum and HCBR classifiers compared with those obtained with SVM and MDM classifiers.

As Figure 10 illustrates, the accuracy of the QSVC (yellow) was a near-perfect circle close to 0.5 and with low inter-subject variation. This indicated that the QSVC failed at the classification task. The diagram also indicates that HCBR (light blue) and the VQC (deep blue) obtained accuracies close to 0.5 but with slightly better results for some of the subjects.

## 6. Discussion and Conclusions

The preliminary results presented in this paper suggest that quantum classification and HCBR are not adapted to the direct classification of EEG signals for P300-based BCIs without preprocessed data. All classifiers did not stand up to a comparison with state-of-the-art classification methods—namely the SVM and MDM classifiers—and obtained a prediction score below the significance threshold of the dataset. However, due to the computation time, we used an exploratory subject to determine the best hyperparameters. It is possible that a systematic approach would have led to better results. In addition, the training and prediction times were significantly higher for all classifiers tested in the current paper compared with those of the state-of-the-art classifiers. To date, the proposed classification techniques should be considered for offline analysis only.

Note that the training accuracies of the QSVC and HCBR were high, but in the case of HCBR, the results may have indicated that the training was too specific to the training data to be useful during prediction. In other words, HCBR likely requires a larger number of samples to achieve a higher prediction accuracy. Since EEG recordings are only valid for a single session and subject, the number of samples that can be collected during an experimental session is limited unless a transfer learning approach is used [62].

However, HCBR performed better with preprocessed data. This was in fact in line with a previous study that demonstrated the suitability of classification methods based on Riemannian geometry for the classification of EEG signals with covariance matrices [17].

Regarding the performance obtained with quantum classification, it is possible that the number of shots was too low with regard to the feature dimension (i.e., [10]). In fact, a high feature dimension increases the number of possible measurements; that is, the number of possible mistakes. A second reason could be that the kernel in either the VQC or QSVC was too simple to obtain a quantum advantage compared with a classical computer. In practice, this could be improved by changing the settings of the feature map by increasing the depth of the circuit or the entanglement method; here, we only used two repetitions and linearly entangled the data. This also suggests that quantum classification would work more effectively in situations in which a classical SVM classifier has failed, which may be the case in experiments with people with BCI illiteracy. In fact, only one dataset was considered in the present study. Further analyses with different datasets are required to reach a broader consensus. Another possible improvement to the VQC could be to increase the depth of the parametrized circuit while also considering that a trade-off exists between accuracy improvement and quantum decoherence [40]. Finally, it is possible to increase the number of filters to select a higher number of relevant components under the conditions that the dimension of the resulting vector is reduced because the capability of the quantum backend is limited.

Another research direction is to investigate the use of quantum computing applied to the MDM classifier. Two studies have supported this suggestion. The first presented the Manifold Convex Class Model [63], which is an effective approach for transforming SPD manifold classes into a convex model, and ran the classification by computing the distances to the convex models. Second, in [64], the authors demonstrated that quantum computing applied to convex optimization problems can result in additional acceleration compared with classical computing, which in turn may allow researchers to diminish the risk of error by lowering the decision threshold.

Therefore, rather than a definite no-go, these results should be considered a first step toward further research on quantum classification and metric learning applied to EEG data.

## Figures and Tables

**Figure 1 brainsci-13-00303-f001:**
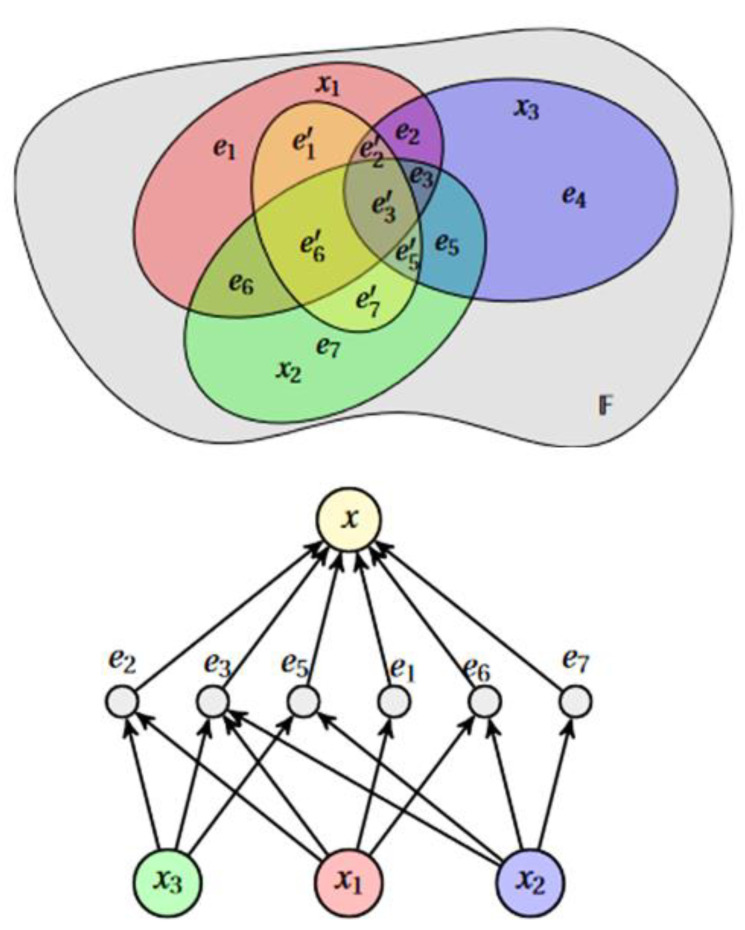
HCBR models a dataset as a hypergraph in which each vector (case) is an edge. The partition of cases forms a tribe on which probabilistic modeling is based. $x_{i}$ denotes a case and $e_{i}\;$ one element of the case partition.

**Figure 2 brainsci-13-00303-f002:**
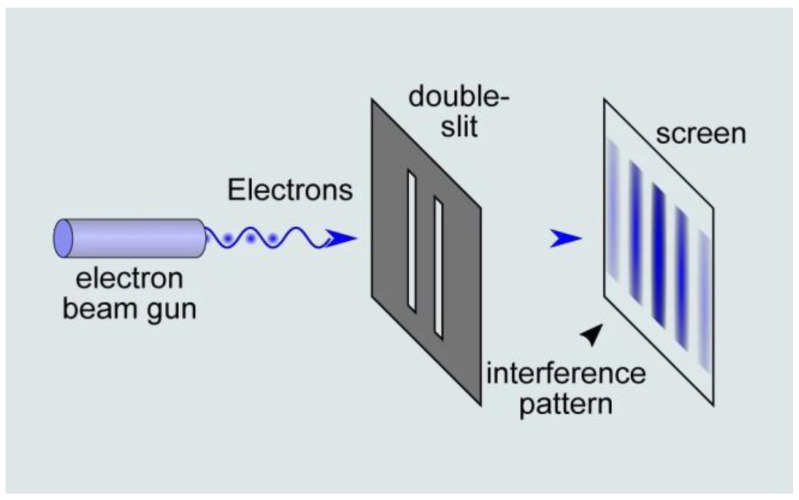
The double-slit experiment illustrates the wave–paticle duality of light. An electron beam consisting of one or multiple electrons is presented in front of a double slit, which produces an interference pattern. Although the electron is a particle, it behaves as a wave in this situation. (source: https://naturenoon.com/double-slit-experiment-simple, accessed on 20 December 2022).

**Figure 3 brainsci-13-00303-f003:**
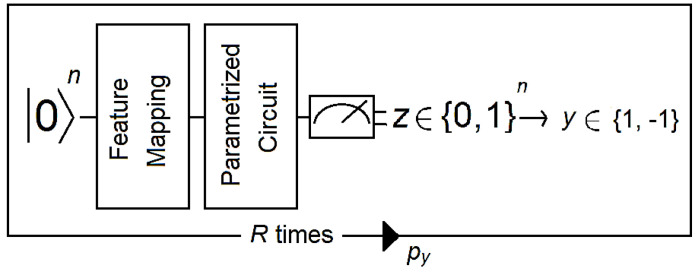
The circuit takes a reference quantum state and applies a feature map followed by a parametrized circuit. Then, the output of the parametrized circuit is measured, and the resulting bit string z is mapped to a label y (1 or −1 in this example). A distribution is generated by running the circuit R times. Then, the expectation value *p*_y_ is estimated for all labels, from which the final label is deduced. Figure adapted from [40].

**Figure 4 brainsci-13-00303-f004:**
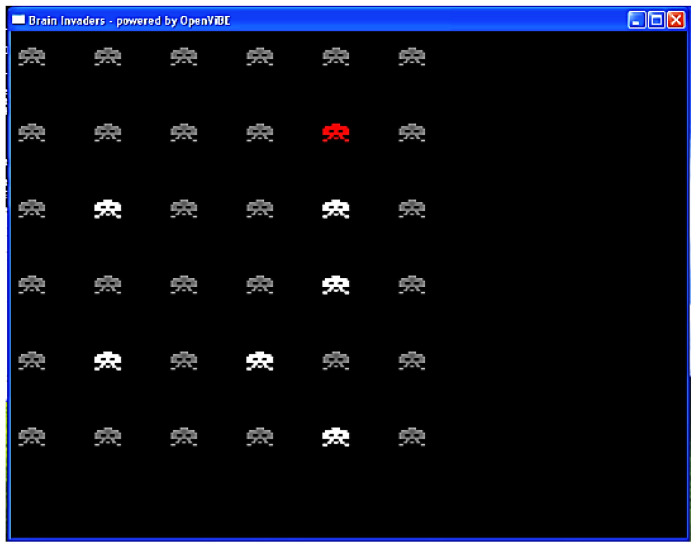
Interface of Brain Invaders at the moment at which a group of six NON-TARGET symbols flashed (in white). The red symbol is the TARGET. The NON-TARGETs that did not flash are in gray. Figure reused from [50].

**Figure 5 brainsci-13-00303-f005:**
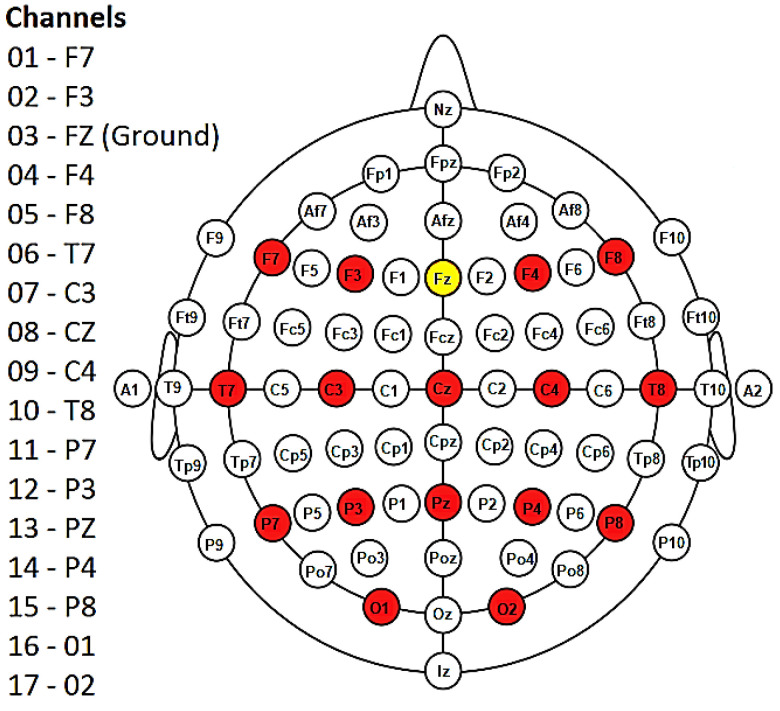
Diagram showing the 16 electrodes (in red) placed according to the 10–20 international system. Fz (in yellow) represents the ground. Note that the NeXus-32 headset does not use an electrode as a reference; rather, a hardware common average reference is used. Figure modified from [50].

**Figure 6 brainsci-13-00303-f006:**
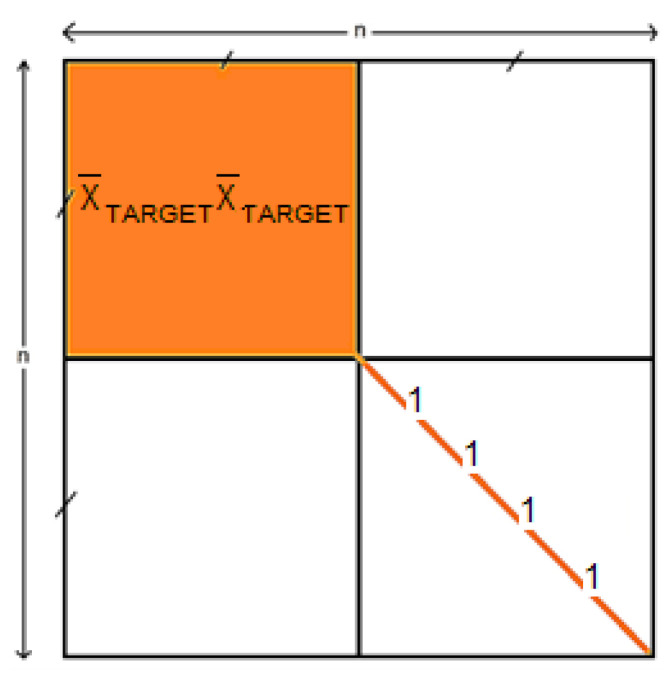
Schema of a correlation matrix of size *n*. Information that was removed (resp. retained) from the correlation matrix before being vectorized is highlighted in orange (resp. white).

**Figure 7 brainsci-13-00303-f007:**
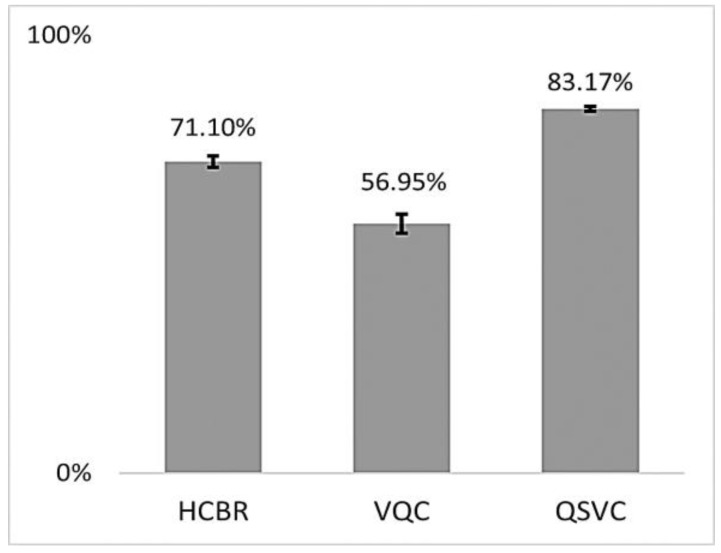
Mean balanced accuracies (in %) of HCBR, VQC, and QSVC achieved during training for all subjects. The vertical black bars show the standard deviation.

**Figure 8 brainsci-13-00303-f008:**
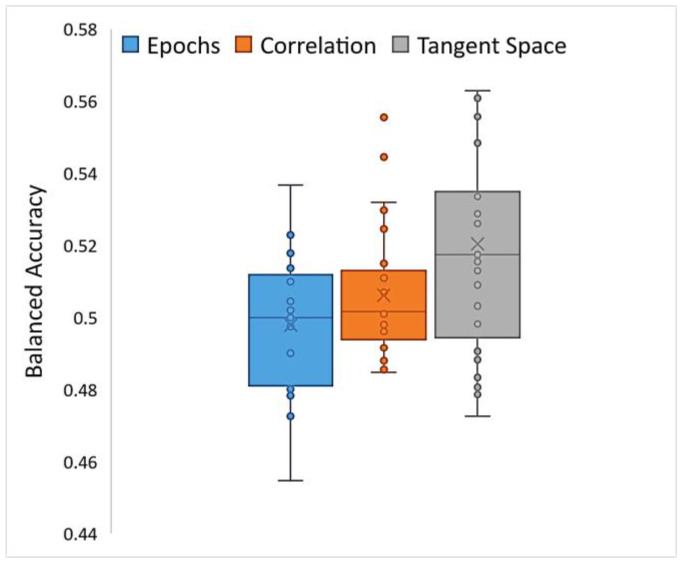
Box plot of balanced accuracies obtained while running HCBR with epoch vectors, correlation vectors, or correlation matrices projected into the tangent space.

**Figure 9 brainsci-13-00303-f009:**
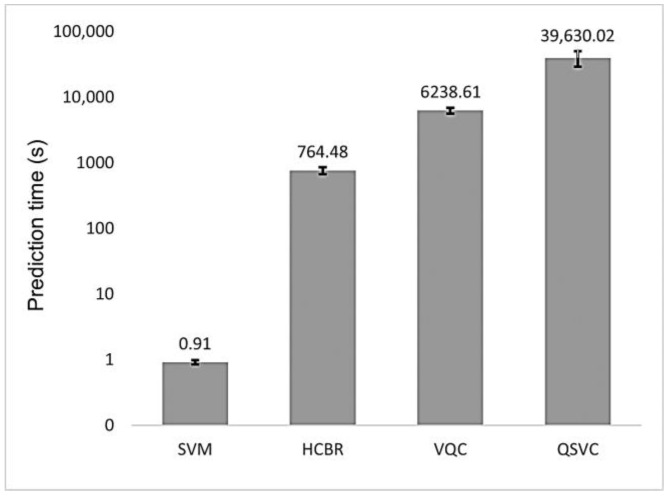
Mean prediction time (in seconds) for HCBR (764.48), VQC (6238.61), and QSVC (39,630.02) as compared to the SVM (0.91) classifier. For the quantum classifiers, we reported the prediction time obtained with our local machine. The vertical scale is logarithmic. The vertical black bars on top of each “column” represent the standard deviation (std) of the classifier: SVM (0.07), HCBR (93.20), VQC (613.58), and QSVC (10,736.88).

**Figure 10 brainsci-13-00303-f010:**
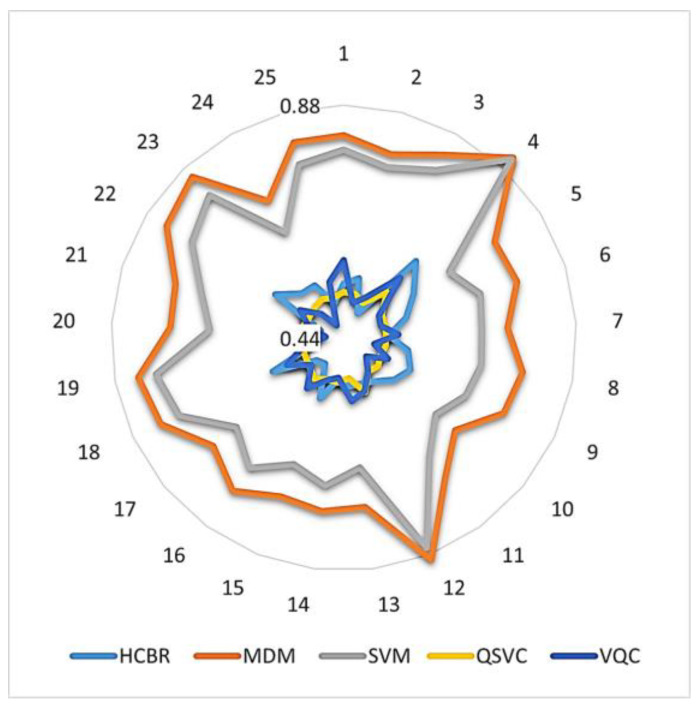
Intra-subject balanced accuracy of quantum and HCBR vs. SVM and MDM classifiers. The numbers around the circle indicate each participant. Accuracies are displayed on a logarithmic scale from 0.44 to 0.88.

**Table 1 brainsci-13-00303-t001:** HCBR-balanced accuracy (PREDICTION) for a single subject depending on the number of decimal digits and xDAWN filters ^a^.

		Number of Decimal Digits		
		4	6	8
**Number of filters**	**1**	0.5367	0.5087	0.4976
	**2**	0.4914	0.4625	0.5030
	**4**	0.4950	0.4567	0.4982

^a^ Scores are highlighted using a color scale from yellow (lowest) to green (highest).

## Data Availability

We used data recorded at GIPSA-lab (Saint-Martin-d’Hères, France) that are freely available on Zenodo (Geneve, Switzerland) at https://zenodo.org/record/2649069 (accessed on 20 December 2022).

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
