# Peer review of "Case-Based and Quantum Classification for ERP-Based Brain–Computer Interfaces"

_brainsci, 2023, doi:10.3390/brainsci13020303_

Round 1
Reviewer 1 Report
The paper is nicely written and seems to be well structured. I have only a few commment for the authors as follows:
1. Please compute ITR metric for your analysis. You may use the following paper as a potential reference (https://ieeexplore.ieee.org/document/9484745).
2. please also show the complexity and required time for running your algorithm.
Author Response
We are grateful to the reviewers for their comments and suggestions, which we have found very useful to improve the manuscript. We have amended the manuscript according to the reviewers’ requests and answered all comments, as reported here below. In addition, we also included supplementary considerations in our discussion. The modification we made are outlined in yellow in the current revision of the article.

Reviewer 2 Report
In this study, the authors investigated the performance of variational quantum, quantum-enhanced support vector, and hypergraph case-based reasoning classifiers, on the binary classification of EEG data from a P300-experiment. The authors found that the balanced training (prediction) accuracy of these three classifiers was 56.95 (51.83), 83.17 (50.25), and 71.10% (52.04%), respectively. In addition, case-based reasoning performed significantly lower with a simplified (49.78%) pre-processing pipeline. These results demonstrated that all classifiers were able to learn from the data and that quantum classification of EEG data was implementable; however, more research is required to enable greater prediction accuracy as none of the classifiers were able to generalize from the data. This could be achieved through improving the configuration of quantum classifiers and increasing the number of trials for hypergraph case-based reasoning classifiers through transfer learning. This is an interesting study, but there are still some problems that cause my concern.
1、The paper should use more BCI data sets for experimental verification.
2、Some figures in the paper are not very clear. Caption contains very little information.
3、Some EEG-related papers seem to be lacking in recent years, such as [1,2]. The paper should add more discussion in the introduction or related work.
[1] Multi-modal Physiological Signals based Squeeze-and-Excitation Network with Domain Adversarial Learning for Sleep Staging[J]. IEEE Sensors Journal, 2022.
[2]Hybrid spiking neural network for sleep electroencephalogram signals[J]. Science China Information Sciences, 2022, 65(4): 140403.
Author Response

(The authors gave the same response as above.)

Round 2
Reviewer 2 Report
The authors have solved all my problems.